# Induction of Immunological Antitumor Effects by the Combination of Adenovirus-Mediated Gene Transfer of B7-1 and Anti-Programmed Cell Death-1 Antibody in a Murine Squamous Cell Carcinoma Model

**DOI:** 10.3390/cancers16071359

**Published:** 2024-03-30

**Authors:** Makiko Hara, Sumiyo Saburi, Natsumi Uehara, Takahiro Tsujikawa, Mie Kubo, Tatsuya Furukawa, Masanori Teshima, Hirotaka Shinomiya, Shigeru Hirano, Ken-ichi Nibu

**Affiliations:** 1Department of Otolaryngology-Head and Neck Surgery, Kobe University Graduate School of Medicine, Kobe 650-0017, Japan; mn.as.you.like@gmail.com (M.H.); kubo@med.kobe-u.ac.jp (M.K.); ftatsuya@med.kobe-u.ac.jp (T.F.); mateshim@med.kobe-u.ac.jp (M.T.); hshino@med.kobe-u.ac.jp (H.S.); nibu@med.kobe-u.ac.jp (K.-i.N.); 2Department of Otolaryngology-Head and Neck Surgery, Kyoto Prefectural University of Medicine, Kyoto 602-8566, Japan; stomita@koto.kpu-m.ac.jp (S.S.); tu-ji@koto.kpu-m.ac.jp (T.T.); hirano@koto.kpu-m.ac.jp (S.H.)

**Keywords:** tumor microenvironment, CD80, CD8, PD-L1, PD-1

## Abstract

**Simple Summary:**

Head and neck cancer is the seventh most common cancer and most cases of head and neck cancer are squamous cell carcinoma (SCC). Recently, immune checkpoint inhibitors (ICPIs) such as anti-programmed cell death-1 (PD-1) have been developed. The aim of our study was to evaluate the antitumor effect of adenoviral vector carrying B7-1 (AdB7) in a murine SCC model in order to further explore the potential of the B7-1 gene in immunotherapy for head and neck cancers. Results indicated that tumor size was significantly reduced in the mice treated with both AdB7 and anti-mouse PD-1 antibody. Additionally, treatment resulted in significantly increased cell densities of total immune cells and Ki-67+ CD8+ T cells and decreased the number of regulatory T cells. Our findings indicate that adenovirus-mediated B7-1 gene expression may enhance the antitumor effect of antiPD1 against SCC.

**Abstract:**

Background: The goal of this study was to evaluate the antitumor immune effects of B7-1 gene expression in addition to immune checkpoint inhibitor against squamous cell carcinoma. Methods: A murine SCC cell line, KLN205, was infected with adenoviral vector carrying B7-1 (AdB7). Infected cells were injected subcutaneously in the flanks of DBA/2 mice. Three weeks after implantation, anti-mouse PD-1 antibody (antiPD1) was intraperitonially administrated twice a week for a total of six times. Results: CD80 was significantly overexpressed in the AdB7-infected tumors. IFN-gamma in the T cells in the spleen was significantly increased and tumor size was significantly reduced in the mice treated with both AdB7 and antiPD1. Targeted tumors treated with both AdB7 and antiPD1 exhibited significantly increased cell densities of total immune cells as well as Ki-67+ CD8+ T cells and decreased regulatory T cells. Conclusions: These results suggest that the B7-1 gene transfer may enhance the antitumor effect of anti-PD1 antibody against SCC.

## 1. Introduction

Head and neck cancer is the seventh most common cancer, accounting for an estimated 888,000 new cases in 2018 worldwide [1]. Most head and neck cancers are squamous cell carcinoma (SCC) of the upper aerodigestive tract [1]. Despite significant advances in diagnostic imaging, surgical techniques, induction chemotherapy [2], and concurrent chemoradiotherapy [3], oncological and functional outcomes of the patients with advanced head and neck SCC have not improved over the past two decades [4].

The latest breakthroughs in comprehending the immune system’s response to cancer have culminated in the innovation of immune checkpoint inhibitors (ICPIs), such as antibodies against programmed cell death-1 (PD-1) [5,6], programmed cell death-ligand 1 (PD-L1) [7], and cytotoxic T-lymphocyte-associated protein 4 (CTLA-4) [8]. Demonstrated to be efficacious in treating not just malignant melanoma but also a wide spectrum of other cancers, including head and neck cancers, these ICPIs represent a significant advancement in cancer therapy. Now, anti-PD-1 antibodies (antiPD1) nivolumab5 and pembrolizumab6 have been approved for the treatment of head and neck cancers and have provided long-term survival to patients with locally advanced and metastatic head and neck cancers. However, at present, these benefits are seen in a limited number of patients. Thus, there is an urgent need for new therapeutic strategies to enhance the oncological effect of ICPIs to conquer these challenging diseases.

T lymphocytes are deemed essential for antitumor immunity [9]. The activation of primary CD8+ cytolytic T lymphocytes (CTLs) is initiated by the recognition of tumor-associated antigens in conjunction with major histocompatibility complex (MHC) class I. Moreover, the activation and induction of CTLs necessitate additional costimulatory signals. Direct presentation of tumor-associated antigen by tumor cells in the absence of co-stimulation not only fails to activate T lymphocytes but may even induce “clonal anergy” [10].

Within the array of costimulatory molecules, the B7 family stands out as particularly influential, with B7-1 (CD80) recognized as the first identified member [11]. B7-1 interacts with CD28 and CTLA-4, which are counterreceptors on T lymphocytes, playing a crucial role in their activation process. Predominantly found on antigen-presenting cells including dendritic cells, activated macrophages, and B cells, B7-1, however, is rarely expressed on most solid tumor cells [11,12]. Low or negative expression of B7-1 in tumor cells is thought to provide an opportunity for escape from the antitumor immune system [12,13]. Notably, our previous study [14] and several other studies have demonstrated the potential of enhancing antitumor responses through the transfection of the B7-1 gene into tumor cells [15,16,17].

In the present study, we evaluated the antitumor effect of B7-1 gene expression in a murine SCC model by combination B7-1 mediated by an adenoviral vector and anti-mouse PD-1 antibody. We demonstrated the significant synergetic effect of B7 gene expression with anti-PD-1 against murine SCC.

## 2. Materials and Methods

### 2.1. Animals and Cell Line

Male DBA/2 mice were purchased from The Jackson Laboratory Japan Inc. (Yokohama, Japan). Mice were 6 weeks of age at the time of the experiment. All the procedures were approved by the Committee for Animal Experiments of Kobe University School of Medicine and performed under the Guidelines for Animal Experiments. KLN205, which was obtained from Riken Bioresource Center, is an established cell line of SCC generated from a DBA/2 mouse. KLN205 cells were cultured in minimum essential medium (MEM) containing 10% fetal bovine serum, penicillin G potassium, and 0.1 mM non-essential amino acids. They were maintained at 37 °C in a 5% CO_2_ humidified incubator. Cells were passaged two times a week into fresh growth medium.

### 2.2. Adenoviral Vectors

We acquired an adenoviral vector containing the B7-1 gene, designated as AxCAmB7:AdB7, from the RIKEN BioResource Research Center (Tsukuba, Japan). This vector, AdB7, is essentially a genetically engineered adenovirus that has been modified to express the B7-1 (CD80) gene, which is commonly found in mice. The genetic expression of this virus is under the control of a hybrid promoter composed of elements from the cytomegalovirus enhancer and the chicken β-actin promoter, often referred to as the CAG promoter. In order to precisely measure the concentration of this viral stock, a well-established plaque-forming assay was employed, utilizing 293 cells, in accordance with methodologies detailed in previous scientific literature (specifically [14]). This particular research undertaking received the official sanction from the Committee for Safe Handling of Living Modified Organisms at Kobe University, with all experimental procedures adhering strictly to the stipulated guidelines of said committee, thereby ensuring compliance with established safety and ethical standards.

### 2.3. Detection of B7-1 Expression on KLN205 Cells Infected with AdB7

We investigated the expression of B7-1 in KLN205 cells infected with AdB7 at various concentrations (multiplicity of infections (MOIs) of 0, 1, and 5) and incubated for 96 h. Total RNA was extracted by TRIzol (Invitrogen, Eugene, OR, USA). Expression of B7 mRNA was analyzed by real-time polymerase chain reaction (RT-PCR) according to the manufacturer’s protocol. Beta-actin mRNAs were used as internal controls. The primer sequences to amplify each gene were as follows [18,19]: B7-1 forward, 5′-TGTATGCCCAGGAAACAGGT-3′ and reverse 5′-TAATCAGTGGTGRTCGGGCT-3′: β-actin forward primer 5′-TCTTGGCTATGGAATCCTG-3′; reverse primer 5′-GTGTTGGCATAGAGGTCT-3′.

#### Determination of Optimal Dose for AdB7 Infection

The KLN205 cell line was carefully cultured at a density of 0.8 × 10^5^ cells per well in a six-well tissue culture plate according to established protocols. These cells were then infected with the AdB7 adenoviral vector. Infection was carried out at two different doses, with one set of wells receiving at MOIs of 1 and another set receiving at MOIs of 5. This was done to determine the effect of different viral concentrations on cell proliferation and viability. After introducing the virus into the cells, the culture plates were placed in an incubator at a temperature of 37 °C. The cells were incubated under these conditions for 96 h. At the end of this incubation period, a cell count was performed by trypan blue exclusion after harvest to identify and quantify the number of living cells remaining. The numbers of cells in five fields were counted under the microscope.

### 2.4. In Vivo Tumor Studies

To study the effect of AdB7 and its synergistic effect with anti-mouse PD-1 antibody on tumor growth in vivo, we performed in vivo tumor model studies. KLN205 cells (5 × 10^5^) infected with AdB7 at MOIs of 5 suspended in 200 μL MEM were subcutaneously injected into the flanks of male 6-week-old DBA/2 mice with a 27-gauge needle as previously described [14] (AdB7-infected group). The non-infected group was injected with non-infected KLN205 cells (parental cells) in the same manner.

Three weeks after implantation, anti-mouse PD-1 antibody (antiPD1) (InVivoPlusTM anti-mouse PD-1 (CD279) (#BP0273, BioCell, NH, USA) was intraperitonially administrated twice a week for a total of six times (antiPD1 and AdB7 + antiPD1 group).

The largest diameters of the implanted tumors (width × length) were periodically measured in millimeters with a caliper. PBS was intraperitonially administrated as negative control in the same manner. Mice were killed 4 weeks after implantation. Tumors were resected and were stored at −80 °C until immunohistochemical analysis. Spleens were also harvested from the mice at the time of sacrifice, and T cells were extracted from the spleens. Concentrations of interferon-gamma (IFN-γ) in the extracted T cells were measured using Mouse IFN-gamma Quantikine ELISA Kit (Funakoshi, Tokyo, Japan).

### 2.5. Immunohistochemical Analysis

Sections were cut to a thickness of 4 μm and processed as previously described [14]. The monoclonal antibody against mouse CD80 (MAB740; R&D Systems, Inc., Minneapolis, MN, USA) was used at a dilution of the manufacturer’s recommendation. Slides were incubated with the respective antibodies at 4 °C overnight. Vectastain ABC kit Elite (Vector Laboratories, Burlingame, CA, USA) was used for visualization. Diaminobenzidine was used for coloration, and nuclei were counterstained with hematoxylin.

### 2.6. Multiplex Immunohistochemical Analysis

Sections (4 μm) were obtained from FFPE samples. Multiplex IHC was performed as described previously [18]. Briefly, FFPE tumor sections were subjected to protein blocking with a blocking buffer (5.0% goat serum, 2.5% BSA, and PBS) for 15 min, followed by sequential immunodetection with primary and horseradish peroxidase (HRP) polymer-conjugated secondary antibodies (Table 1). Following the removal of chromogenic stains via an alcohol gradient, the tissue slides were subjected to heat-mediated antibody stripping in a citrate buffer (pH 6.0), allowing iterative cycles of immunostaining and visualization. The antibodies utilized and the corresponding staining parameters are shown in Table 1. The digital image processing was performed in accordance with previously established methodologies [20]). Co-registration of the digitized images capturing the antibody panel were performed by in-house software developed by SCREEN Holdings Co., Ltd. (Kyoto, Japan). A sequential gating strategy of image cytometry was applied in order to identify tumor cell phenotypes based on negative cell staining by using FCS Express 7 Plus Version 7.16.0035 (De Novo Software, Pasadena, CA, USA). Tumor and immune cells were identified based on the criteria identification markers in Table 2.

## 3. Results

### 3.1. Detection of B7-1 Expression in KLN205 Cells Infected with AdB7

In the experiments conducted with KLN205 cells, a direct relationship was observed between the amount of B7-1 protein expressed and the concentration of AdB7 adenoviral vector administered, as shown in Figure 1a. This observation suggested that the greater the amount of vector present, the greater the expression of B7-1 within the cells. Subsequent investigations focused on determining the most effective concentration of AdB7 for use with KLN205 cells. The data presented in Figure 1b show that cell proliferation was suppressed when cells were infected with AdB7 at a MOI of 5. Conversely, it was also observed that the level of B7-1 gene expression at an MOI of 5 was significantly increased when compared to the gene expression at an MOI of 1. Taking these findings into account, an MOI of 5 was determined to be the most appropriate concentration for infecting the cells, effectively inducing maximal gene expression of B7-1 while mitigating cytotoxic side effects. After determining the optimal viral dose, the cells were exposed to AdB7 at an MOI of 5. After 72 h of incubation at a controlled temperature of 37 °C, the treated cells were carefully harvested. These harvested cells, expressing high levels of the B7-1 gene and exhibiting minimal cytotoxicity, were then prepared for implantation into mice in order to facilitate further in vivo experimental research.

### 3.2. Antitumor Effects of B7-1 and Anti-PD-1 Antibody

To evaluate the synergistic antitumor effect of B7-1 gene transfer and antiPD1 antibody against mouse SCC, 3 weeks after the transplantation, antiPD1 was intraperitonially injected twice a week for total of six times. As shown in Figure 2, the growth of tumors treated with both AdB7 and antiPD1 was significantly inhibited compared with tumors treated with either AdB7 or antiPD1.

### 3.3. Tumor Microenvironment

IFN-gamma in the T cells in the spleen was significantly increased in the mice treated with both AdB7 and antiPD1. Positive staining for CD80 encoded by B7-1 gene was observed in tumors in the AdB7-infected group but only in occasional cells in non-infected groups (Figure 3). IFN-gamma in the T cells in the spleen was significantly increased in the mice treated with both AdB7 and antiPD1 (Figure 4).

Multiplexed immunohistochemical analysis showed that tumors treated by combination of AdB7 and antiPD1 had higher densities of tumor-infiltrating immune cells (Figure 5a,b(A) and Appendix A). Among immune cells, tumors treated by the combination exhibited relatively higher densities of CD8+ T cells and helper T cells with significantly lower densities of regulatory T cells (Figure 5b(B–D) and Appendix A). Notably, the Ki-67 expression of tumor-infiltrating CD8+ T cells was significantly high in the combination group (Figure 5b(E) and Appendix A), suggesting the enhanced proliferation of T cells.

## 4. Discussion

In the field of preclinical HNSCC research, in vitro models have proven invaluable in providing insights into cellular responses and drug interactions in a controlled laboratory environment. However, to gain a comprehensive understanding of tumor microenvironment (TME), such as the intricate mechanisms and molecular events that occur during the initiation and progression of HNSCC [21], it is imperative to use in vivo animal models. These models are critical because they provide a more dynamic and complex environment that closely mimics the actual conditions within a living organism. This natural habitat is crucial for observing the interaction of cancer cells with their surrounding biological environment, including the immune system, which can significantly influence tumor growth and response to treatment. Therefore, while in vitro studies provide the basis for identifying potential therapeutic targets and testing hypotheses, in vivo animal models are essential for validating these findings and exploring the full spectrum of cancer biology within an organism’s unique biological context [22,23]. In this study, the therapeutic effect of Ad-B7 was investigated by conducting in vitro experiments using SCC cells from DBA/2 mice and further in vivo experiments using DBA/2 mice to reproduce the TME.

The critical role of B7-1 as a co-stimulatory factor in generating an antitumor immune response is suggested by numerous in vivo experimental models, including those involving melanomas, adenocarcinomas, and gliomas [17,24,25,26,27]. Other studies have demonstrated that immunizing mice with B7-1–transfected tumor cells elicits protective and even curative immunity against wild-type tumors [28]. Accordingly, in our previous study, we demonstrated the antitumor effect of the B7-1 gene mediated by adenovirus and showed the potential of the B7-1 gene for tumor immunotherapy against SCC [14].

Encouraged with these findings, in this study, we evaluated the synergistic immunological antitumor effect of B7-1 gene expression and antiPD1 against SCC using a murine SCC Model. As expected, the growth of tumors treated with both AdB7 and antiPD1 was significantly inhibited in size compared with tumors treated with either AdB7 or antiPD1 alone. In the tumors treated with both AdB7 and antiPD1, IFN-gamma was significantly increased in the T cells. Notably, the number of cytotoxic and helper T cells as well as tumor-associated macrophages were significantly increased, while helper T cells were significantly decreased. These results suggest that the combination of B7-1 gene expression in the targeted tumor and systemic administration of antiPD1 had synergically enhanced the antitumor immunity against SCC.

One thing to consider is that B7-1 (CD80) binds not only to CD28, but also to CTLA-4 [11]. When B7-1 binds to CD28, B7-1 delivers stimulation signals to the immune response and induces the survival of T lymphocytes. However, the interaction of B7-1 with CTLA-4 delivers inhibitory signals for activation of T-lymphocytes [29,30,31]. So far, previous studies and our study have demonstrated that the transfection of B7-1 works as a co-stimulatory factor that induces an immunogenicity mechanism to suppress the growth and/or metastasis of the targeted tumors, as above mentioned, which suggests that its effect as a co-stimulatory factor outweighs its effect as a co-inhibitory factor. However, the numbers and types of cell lines used in each study were still limited. Another limitation is the number of cell lines in this study. Since SCC cell lines derived from DBA/2 mice are extremely rare, only one cell line was available for the murine SCC model. Taken together, further studies are needed to reach a more definitive conclusion on the antitumor effect of B7-1 transfection with antiPD1 against SCC.

## 5. Conclusions

The results obtained by this study suggest that adenovirus-mediated B7-1 gene expression may enhance the antitumor effect of antiPD1 against SCC. Further studies are warranted before clinical application.

## Figures and Tables

**Figure 1 cancers-16-01359-f001:**
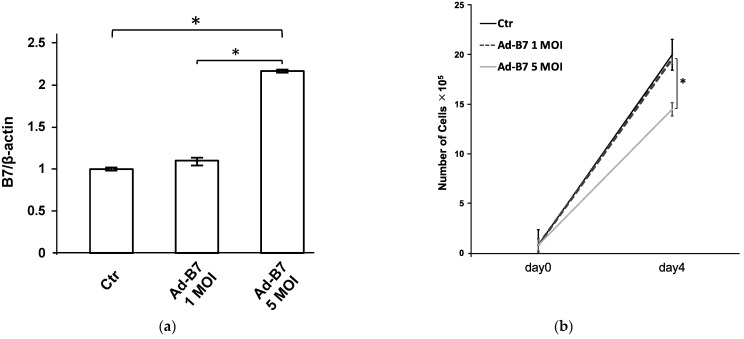
(**a**) B7-1 expression in KN205 cells infected with AdB7. KLN205 cells were infected with AdB7 at various concentrations. Real-time polymerase chain reaction was used for detection of B7-1 expression. The expression increased in proportion to the concentration of vector infection. * *p* < 0.05. Error bars indicate the standard error of the mean. (**b**) Cell proliferation and viability of KLN205 cells infected with AdB7. Proliferation of AdB7-infected KLN205 cells were significantly inhibited at 5 multiplicities of infections (MOIs), while there were no significant differences in proliferation between AdB7-infected KLN205 cells at 1 MOI and non-infected KLN205 cells. * *p* < 0.05. Error bars indicate the standard error of the mean.

**Figure 2 cancers-16-01359-f002:**
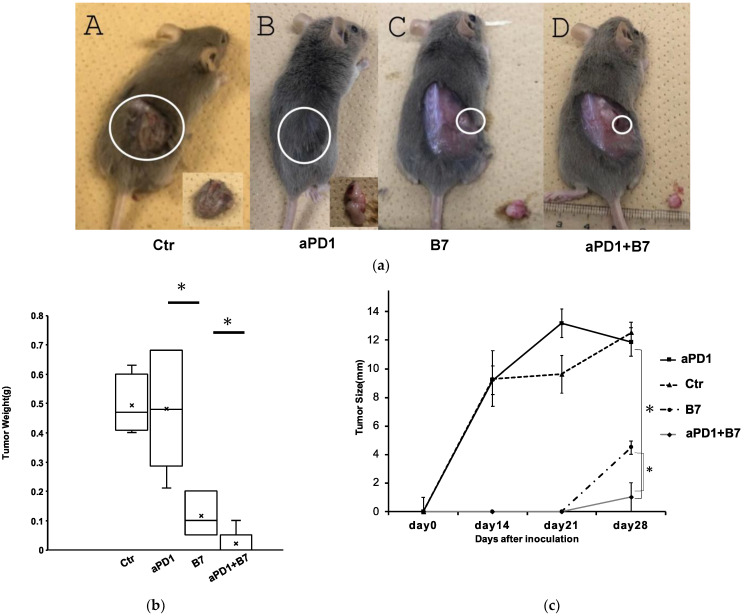
Antitumor effect of B7-1 gene expression and anti-PD-1 antibody. (**a**) Tumors 4 weeks after implantation. AdB7-infected KLN205 cells or non-infected KLN205 cells were subcutaneously injected into the flanks of DBA/2 mice. The combination of B7 gene expression and antiPD-1 antibody significantly inhibited tumor growth. Circles indicated where the tumor was located. (**A**) Control: average weight 0.49 g, (**B**) antiPD-1: average weight: 0.48 g, (**C**) B7: average weight 0.12 g, (**D**) B7 + antiPD-1: average weight 0.02 g. (**b**) Tumor weights 4 weeks after inoculation. (**c**): Diameter of tumors 2 weeks, 3 weeks, and 4 weeks after inoculation. Diameters were determined as the average of the long diameter and short diameter. * *p* < 0.05. Error bars indicate the standard error of the mean. Ctr: Control; aPD1: anti-PD-1 antibody; B7: AdB7.

**Figure 3 cancers-16-01359-f003:**
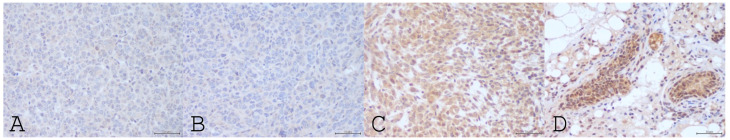
CD8-positive cells in the noninfected group. Photographs of immunohistochemical staining using antibody against CD80. (**A**) Control, (**B**) anti PD1, (**C**) AdB7, (**D**) AdB7 + antiPD1. The majority of tumor cells in the AdB7-infected cells groups strongly expressed CD80 (**C**,**D**), but not in the non-infected groups (**A**,**B**).

**Figure 4 cancers-16-01359-f004:**
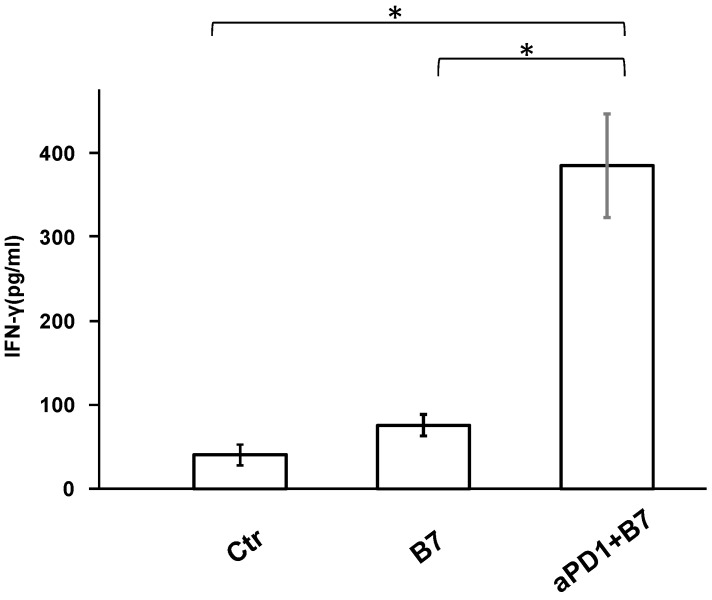
IFN-gamma in the T cells in the spleen. Spleens were harvested from the mice at the time of sacrifice, and T cells were extracted from the spleens. Concentrations of interferon-gamma (IFN-γ) in the extracted T cells were measured using Mouse IFN-gamma Sandwich ELIA KIT (Proteintech^®^, INC). IFN-gamma in the T cells in the spleen was significantly increased in the mice treated with both AdB7 and antiPD1. Ctr: Control; aPD1: anti-PD-1 antibody; B7: AdB7. * *p* < 0.05.

**Figure 5 cancers-16-01359-f005:**
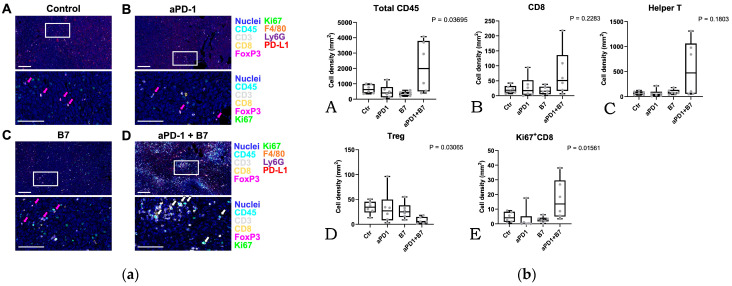
(**a**) Visualization of infiltrating immune cells via 9-marker multiplex immunohistochemistry. Formalin-fixed paraffin-embedded (FFPE) tissue sections derived from control (**A**), anti-PD-1 antibody (aPD1) (**B**), AdB7 (**C**), and aPD1 plus AdB7 groups (**D**) were subjected to 9-marker multiplex immunohistochemistry. Pseudo-colored merged composite images are shown. Boxed regions in the upper panels show magnified areas in the lower panels. Markers and color annotations are shown on the right. White arrows present CD3+ CD8+ Ki67 + cells. Magenta arrows present CD3+ CD8 FoxP3 + cells. Scale bars = 100 μm, Ctr: Control; aPD1: anti-PD-1 antibody; B7: AdB7. (**b**) Tumors treated by combination of AdB7 and anti-PD1 antibody had higher densities of tumor-infiltrating immune cells (**A**). Among immune cells, tumors treated by the combination exhibited relatively higher densities of CD8+ T cells and helper T cells with significantly lower densities of regulatory T cells (**A**–**D**). Notably, the Ki-67 expression of tumor-infiltrating CD8+ T cells was significantly high in the combination group (**E**), suggesting the enhanced proliferation of T cells. Ctr: Control; aPD1: anti-PD-1 antibody; B7: AdB7.

**Table 1 cancers-16-01359-t001:** A complete list of antibodies and conditions used for staining.

	Cycle 1	Cycle 2	Cycle 3	Cycle 4	Cycle 5	Cycle 6	Cycle 7	Cycle 8	Cycle 9
Primary Ab	Hematoxylin	PDL1	CD8	CD3	F4/80	CD45	Foxp3	Ki67	Ly6G
Supplier	Dako	Proteintech	eBioscience	Gene Tex	Abcam	BD Pharmingen	Invitrogen	Cell Signaling	ABA
Clone/Product #number	S330130-2	17952-1-AP	14-0808-82	SP7	ab6640	30-F11	FJK16s	D3B5	1A8
Conc	Original	1/200	1/50	1/400	1/400	1/200	1/100	1/10,000	1/2000
Reaction time	2 min	30 min	30 min	30 min	30 min	30 min	30 min	30 min	30 min
Secondary Ab		Anti-rabbit	Anti-rat	Anti-rabbit	Anti-rat	Anti-rat	Anti-rat	Anti-rabbit	Anti-rat
Reaction		30 min	30 min	30 min	30 min	30 min	30 min	30 min	30 min
AEC		20 min	20 min		20 min	20 min	20 min	20 min	20 min
AMEC				5 min					
Heat treatment	Citrate	Citrate	Citrate	Citrate	Citrate	Citrate	Citrate	Citrate	Citrate

**Table 2 cancers-16-01359-t002:** A list of lineage identification markers.

Cell Type	Identification Biomarkers
CD8^+^ T cells	CD45^+^ CD3^+^ CD8^+^
Proliferating CD8^+^ T cells	CD45^+^ CD3^−^ CD8^−^ Ki-67^+^
Hepler T cells ^#^	CD45^+^ CD3^+^ CD8^−^ FoxP3^−^
Regulatory T cells (Treg)	CD45^+^ CD3^+^ CD8^−^ FoxP3^+^
Tumor-Associated Macrophages	CD45^+^ CD3^−^ CD20^−^ F4/80^+^
Granulocytes	CD45^+^ CD3^−^ CD8^−^ Ly6G^+^

# Due to a lack of multiplex availability for CD4, CD3^+^CD8^−^ cells were considered as CD4+ T cells.

## Data Availability

The datasets generated during and/or analyzed during the current study are available from the corresponding author on reasonable request.

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
