# Peer review of "Induction of Immunological Antitumor Effects by the Combination of Adenovirus-Mediated Gene Transfer of B7-1 and Anti-Programmed Cell Death-1 Antibody in a Murine Squamous Cell Carcinoma Model"

_cancers, 2024, doi:10.3390/cancers16071359_

Round 1
Reviewer 1 Report
Comments and Suggestions for Authors
Hara et al., have evaluated the therapeutic efficacy of the combination of adenovirus-mediated gene transfer of B7-1 and a-PD-1 in a murine squamous cell carcinoma (SCC) model.
KLN205, a murine SCC model was used for this purpose. The results indicated significant increasing of immune cells, and Ki-67+CD8+ T- cells and decreasing of regulatory T-cells in the combination therapy group.
Comments
Figure 2: I can't understand this figure. Usually, vector-infected cell line should show decreased viability. In this figure, y axis shows number of cells, it must be % of viability. Can you explain about this figure.
Line#186: wight=>weight
Figure 5: What about IFN-g level in a-PD-1 group? Please include in a-PD-1 group in this figure.
Figure 6: Please provide clear images.

Author Response
Reviewer 1 comments
Comment 1: Figure 2: I can't understand this figure. Usually, vector-infected cell line should show decreased viability. In this figure, y axis shows number of cells, it must be % of viability. Can you explain about this figure.
Answer 1: Since the purpose of this study was to evaluate the synergic effect of Ad-B7 and anti-PD1, in Figure 2, we aimed to determine the optimum concentration at which proliferation of tumor cell is “not” too much suppressed by AdB7 adenovirus alone. To make it easier for readers to understand the degree of cell proliferation, we determined the vertical axis to show number of cells. According, we changed the title of Figure 2 to “Cell proliferation and viability of KLN205 cells infected AdB7”, to make it easier for readers to understand our intent.
****************************************************************
Comment 2: Line#186: wight=>weight
Answer 2: Thank you for pointing out spelling error. Accordingly, we have corrected the spelling.
****************************************************************
Comment 3: Figure 5: What about IFN-g level in a-PD-1 group? Please include in a-PD-1 group in this figure.
Answer 3: We agree with the reviewer’s advice. IFN-g level in aPD-1 group would be valuable in Figure 5, as the reviewer suggested. According to the rules of our animal facility, the experiment of aPD1 group without AdB7 adenovirus was conducted, in advance, separately from the other groups using AdB7 adenovirus. Unfortunately, we had not come up with the idea of IFN-g experiment at that moment. So, spleens were not harvested in aPD1 group. Therefore, we regret to tell you that it is not possible to add data for the aPD1 group in Figure 4.
(In accordance with the reviewer2's comment, we have merged Figure1,2 to Figure1 and Figure 6,7 to 5. So, Figure 5 changed to Figure 4.)
****************************************************************
Comment 4: Figure 6: Please provide clear images.
Answer 4: We agree that this point requires clarification. We added the Image of each single marker as a Supplementary Figure 1.
Reviewer 2 Report
Comments and Suggestions for Authors
The authors of the present work studied the effect of adenoviral vector carrying B7-1 (AdB7) in a murine SCC model to explore the potential of the B7-1 gene in immunotherapy for HNCs.
The manuscript looks like well written and organized. The authors have presented an interesting topic in the field of HNC treatments. The paper should be considered after major revisions.
1. All the experiments were performed only on one cell line. The authors should test at least another cell line;
2. The authors should use the same software of Figure 7 to present the graphs of Figure 1, 2, 3 and 5 ;
3. Figure 1 and 2 reported the data of in vitro experiments. The authors should merge the two figures. It is the same in Figure 6 and 7;
4. In Figure 6, many markers are difficult to detect. The authors should increase the image dimensions;
5. Well-defined preclinical models are needed to better reproduce the cancer TME. Commercial cell lines cultured on common monolayer supports are in vitro systems that are not able to mimic the microenvironment of cancer diseases. On the other hands, in vivo models represent some valuable research resources to reproduce the TME. For these reasons, the authors should underline these aspects through a short overview of preclinical models developed to reproduce the TME and to underline the importance to integrate different systems in order to increase the complexity of the tool used. The following references should be included in the manuscript: “Preclinical models in head and neck squamous cell carcinoma. doi: 10.1038/s41416-023-02186-1” and “Precision Medicine in Head and Neck Cancers: Genomic and Preclinical Approaches. doi: 10.3390/jpm12060854”.
Author Response
Comment 1: All the experiments were performed only on one cell line. The authors should test at least another cell line
Answer 1: We agree with the reviewer's comment. However, due to the rarity of SCC cell lines derived from DBA/2 mice, we obtained only one cell line for the present study. We hope that the present study provides a valuable information that B7-1 gene expression mediated by adenovirus might enhance the anti-tumor effect of anti-PD1.
***************************************************************
Comment 2: The authors should use the same software of Figure 7 to present the graphs of Figure 1, 2, 3 and 5;
Answer 2: In accordance with the reviewer's comment, we have changed Figures 1, 2, 3, 5.
***************************************************************
Comment 3: Figure 1 and 2 reported the data of in vitro experiments. The authors should merge the two figures. It is the same in Figure 6 and 7;
Answer 3: In accordance with the reviewer's comment, we have merged Figure1,2 to Figure1 and Figure 6,7 to 5.
***************************************************************
Comment 4: In Figure 6, many markers are difficult to detect. The authors should increase the image dimensions;
Answer 4: We agree that this point requires clarification. We added the Image of each single marker as a Supplementary Figure 1.
***************************************************************
Comment 5. Well-defined preclinical models are needed to better reproduce the cancer TME. Commercial cell lines cultured on common monolayer supports are in vitro systems that are not able to mimic the microenvironment of cancer diseases. On the other hands, in vivo models represent some valuable research resources to reproduce the TME. For these reasons, the authors should underline these aspects through a short overview of preclinical models developed to reproduce the TME and to underline the importance to integrate different systems in order to increase the complexity of the tool used. The following references should be included in the manuscript: “Preclinical models in head and neck squamous cell carcinoma. doi: 10.1038/s41416-023-02186-1” and “Precision Medicine in Head and Neck Cancers: Genomic and Preclinical Approaches. doi: 10.3390/jpm12060854”.
Answer 5: Thank you for your advice. We have added the following text to the Discussion (p. 10, line 274-289) and References #21-23.
“In the field of preclinical HNSCC research, in vitro models have proven invaluable in providing insights into cellular responses and drug interactions in a controlled laboratory environment. However, to gain a comprehensive understanding of tumor microenvironment (TME) such as the intricate mechanisms and molecular events that occur during the initiation and progression of HNSCC [21], it is imperative to use in vivo animal models. These models are critical because they provide a more dynamic and complex environment that closely mimics the actual conditions within a living organism. This natural habitat is crucial for observing the interaction of cancer cells with their surrounding biological environment, including the immune system, which can significantly influence tumor growth and response to treatment. Therefore, while in vitro studies provide the basis for identifying potential therapeutic targets and testing hypotheses, in vivo animal models are essential for validating these findings and exploring the full spectrum of cancer biology within an organism's unique biological context [22, 23]. In this study, the therapeutic effect of Ad-B7 was investigated by conducting vitro experiments using SCC cells from DBA/2 mice and further vivo experiments using DBA/2 mice to reproduce the TME.”
Round 2
Reviewer 2 Report
Comments and Suggestions for Authors
Now, the manuscript is acceptable for the pubblication on "Cancers" journal